# Commercial milk formula feeding among children under two years in Nepal: Trends and determinants from four Nepal Demographic and Health Surveys (2006–2022)

**Barun Kumar Singh**[1,2,3]*, **Sangita Bista**[4], **Sajjan Yogesh**[4], **Vishnu Khanal**[5,6]

**1** Health Nutrition Education and Agriculture Research Development, Saptari, Nepal, **2** School of Public Health and Preventive Medicine, Monash University, Melbourne, Australia, **3** Global Women's and Newborn's Health Group, Burnet Institute, Melbourne, Australia, **4** Independent Public Health Consultant, Kathmandu, Nepal, **5** Remote Health Systems and Climate Change Centre, Menzies School of Health Research, Charles Darwin University, Alice Springs, Northern Territory, Australia, **6** Nepal Development Society, Bharatpur, Nepal

* barun272@gmail.com

## Abstract

### Background

Exclusive breastfeeding in the first six months of life and continuation of breastfeeding until the second year are recommended for the healthy growth and development of infants and young children. However, the use of commercial milk formula has increased in recent decades in low and middle-income countries, including Nepal. Understanding the trends and determinants is crucial for designing evidence-based and context-specific infant and young child feeding support programs. There is limited evidence on the use of commercial milk formula in Nepal. Therefore, this study aimed to explore the trends and prevalence of commercial milk formula feeding practices and their associated factors among mothers of children aged 0–23 months in Nepal.

### Method

We used data from four consecutive nationally representative surveys (2006, 2011, 2016, and 2022). A total weighted sample of 7,705 (0–5 months: 1,978; 6–23 months: 5,727) children aged 0–23 months was included in the analysis. The trends were presented as frequency distributions. Multivariable logistic regression was performed to assess the association between individual and underlying factors with commercial milk formula feeding in the first two years of life. For enhanced analytical clarity and programmatic relevance, we stratified our analysis into two distinct age groups: 0–5 months and 6–23 months.

**Data availability statement:** The data used in this study are publicly available from the Demographic and Health Surveys (DHS) Program and can be accessed via the following link: https://dhsprogram.com/data/available-datasets.cfm.

**Funding:** The author(s) received no specific funding for this work.

**Competing interests:** The authors have declared that no competing interests exist.

## Results

Over the study period (2006–2022), the prevalence of commercial milk formula feeding among infants aged 0–5 months increased steadily from 1.95% (95% confidence interval (CI): 0.83–4.55) to 11.09% (95% CI: 8.01–15.16). A similar but less upward trend was observed among children aged 6–23 months, with prevalence rising from 2.03% (95% CI: 1.26, 3.27) to 6.68% (95% CI: 5.26, 8.45). Among 0–5 months, infants born to mothers with secondary and higher education (adjusted odds ratio (aOR): 12.48; 95% CI: 2.63, 59.14), mothers aged 25–34 years (aOR: 2.29; 95% CI: 1.14, 4.60), and born by caesarean section (aOR: 2.16; 95% CI: 1.01, 4.59) were more likely to receive formula feeding compared to their counterparts. Similarly, among 6–23 months, children who were male (aOR: 1.67; 95% CI: 1.15, 2.42), perceived as small at birth (aOR: 2.76; 95% CI: 1.48, 5.15), first born (aOR: 2.59; 95% CI: 1.02, 6.56), born in health facilities (aOR: 3.18; 95% CI: 1.16, 8.73), and whose fathers had secondary or higher education (aOR: 3.14; 95% CI: 1.22, 8.10) were more likely to receive commercial milk formula.

## Conclusion

The prevalence of commercial milk formula feeding practices has increased during 2006–2022. Evidence-informed, context-specific interventions are essential to halt and reverse commercial milk formula feeding practices and protect, promote, and support breastfeeding and recommended infant feeding practices. A program designed to address this issue should target vulnerable groups and leverage the specific factors identified in this study.

## Background

Infant feeding practices during the first two years are crucial for an infant's healthy growth and development, with lifelong implications for children, mothers, and society [1,2]. Any negative repercussions sustained during this period can lead to poor cognitive development, malnutrition, delayed educational milestones, and reduced economic productivity later in life [1,3]. According to recommendations from the World Health Organisation (WHO) and UNICEF, optimal breastfeeding involves initiating breastfeeding within the first hour after birth and exclusively breastfeeding (EBF) for the first six months. Breastfeeding should continue for up to two years, with complementary foods gradually introduced after six months [1]. The WHO recommends introducing complementary foods between 6 and 8 months and continuing them alongside breastfeeding. However, according to the updated global estimates by UNICEF, less than half (48%) of infants aged 0–5 months were exclusively breastfed globally within the 24 hours preceding the surveys. Similarly, only 59% of children aged 12–23 months continue breastfeeding for two years [4]. WHO and UNICEF expect all member states to achieve 70% EBF rates in the first six months of an infant's life and 60% continuation rates of breastfeeding among children up to two years of age by the end of 2030 [5]. Universal

breastfeeding practices among children under two years of age could prevent an estimated 8,23,000 deaths, accounting for 13.8% of total deaths in low and middle-income countries (LMICs) per year [6]. Additionally, universal breastfeeding practices worldwide could prevent an estimated 20,000 avoidable breast cancer deaths annually worldwide [6].

Evidence suggests that the proliferating corporate marketing of commercial milk formula (CMF) in recent decades, among other factors, has been a pivotal contributor to the decline in exclusive and continued breastfeeding. These marketing strategies have changed the feeding environment for infants and young children, particularly in LMICs [7,8]. CMF products include any milk in liquid or powder form, packaged as the second-best option, and available in multiple categories (standard, follow-up, and toddler formula), often marketed for a healthy child [9,10]. The marketing strategy has influenced social norms and hindered the adoption of breastfeeding practices [7,11], resulting in a greater number of infants being fed formula milk instead of breast milk globally [2]. These marketing efforts have proven to be highly effective investments for formula companies. The global retail sales of CMF expanded by 121.5%, increasing from 0.97 to 2.15 million tonnes from 2005 to 2019 [12]; this growth is expected to continue, with sales projected to rise by another 18.6%, reaching 2.55 million tonnes by 2030 [13], reflecting a situation where more infants and young children are provided CMF than ever [14]. Its global market, valued at approximately USD 68.27 billion in 2022, is projected to exceed USD 174.66 billion by 2032, with a compound annual growth rate of 9.84% from 2023 to 2032 [15]. The most recent growth in formula feeding practices was attributed to upper-middle-income countries, followed by lower-middle-income countries, where growth was also strong, with a 122.3% increment over the same period [12]. Formula feeding practices are strongly and negatively correlated with continued breastfeeding at 12–15 months in LMICs [16].

This notable increase in the use of formula feeding raises concerns for both human and planetary health. It results in the loss of the health, developmental, and food security benefits associated with breastfeeding, and contributes to global warming and other environmental harm [12,16,17]. Formula feeding is associated not only with a higher risk of gastrointestinal and respiratory tract infections [18] but also with an increased risk of non-infectious conditions, including allergic diseases, type 1 and type 2 diabetes [19], and obesity [20]. Formula feeding also has a consistently higher carbon footprint than breastfeeding [21]. Additionally, formula feeding practices cost mothers six times more than exclusive breastfeeding during the first six months [22], resulting in a financial burden on the family. Despite the well-demonstrated evidence of the disadvantages of formula milk, formula feeding practices remain widespread in LMICs, particularly in urban areas [6,16], including Nepal [23].

Only one study has examined the use of CMF in Nepal, and it was limited to a western region of the country [23]. This prospective cohort study demonstrated that 31.7% of urban infants received infant formula within the first six months [23]. Despite the widespread use and poor outcomes associated with formula feeding [24], there is a paucity of evidence from nationally representative studies. Details on factors associated with CMF feeding practices during Nepal's Millennium Development Goal (MDG) and Sustainable Development Goal (SDG) periods (2006−2022) are also lacking. Despite Nepal's strong focus and notable success in reducing under-five and infant mortality during the MDG era (2000−2015), studies on CMF use among vulnerable populations were lacking. Furthermore, the adverse effects of inappropriate preparation and use of formula milk have rarely been the subject of scientific inquiry. This study aimed to investigate the trends and determinants of CMF feeding practices in Nepal between 2006 and 2022. Findings from this study will be highly relevant to public health experts and policy makers in Nepal, given the current SDG target to achieve 90% EBF practices and the WHO/UNICEF recommendation of continued breastfeeding practices until the second year of life. These efforts will ultimately contribute to achieving the SDG-2 agenda, which reflects a strong global political commitment to end malnutrition in all its forms by 2030 [25].

## Method

### Context/study setting

Nepal is one of the LMICs, located in South Asia, with a population of approximately 30 million, and about two-thirds of the population (66.17%) residing in urban areas [26]. According to the Nepal Demographic and Health Survey (NDHS)

2022, the prevalence of EBF among infants under six months was 56%, while the median duration of breastfeeding was 29.5 months [27]. However, the early initiation of breastfeeding (within one hour of birth) remains suboptimal at 49%, and prelacteal feeding (giving liquids or foods other than breast milk in the first three days) was practiced for 26% of newborns [27]. Achieving the WHO-recommended EBF for six months and continuing breastfeeding up to two years remains a significant public health challenge in Nepal. The NDHS 2022 also highlights that infant formula use is rising, with 8% of children under six months receiving formula milk [27]. Notably, births assisted by skilled health professionals or occurring in health facilities are associated with higher rates of prelacteal feeding, possibly due to medicalised practices or lack of immediate breastfeeding support [23].Despite national policies promoting EBF, challenges persist, including low awareness of the risks associated with formula and bottle feeding, as well as cultural practices favouring prelacteal feeds (e.g., honey, water) [23]. Similarly, aggressive marketing of CMF and work-related barriers for employed mothers also exists [28]. While the NDHS provides critical data, it does not fully explain the underlying reasons for suboptimal breastfeeding practices. Potential drivers include socioeconomic disparities, gaps in maternal education, and insufficient counselling during antenatal visits [23,29,30].

Nepal adopted the 1981 "International Code for Marketing Breastmilk Substitutes" of the WHO, developed in response to the global outcry against commercial milk formula, following the controversy over products produced by Nestlé and other manufacturers, which were labelled as "baby killers" in 1977 [31]. The government of Nepal (GoN) enacted the "Mother's Milk Substitutes (Control of Sale and Distribution) Act, 2049 (1992) and the corresponding Regulation, 2051 (1994)", which strictly banned the marketing, advertisement, and promotion of formula milk [32]. In 2014, the GoN also endorsed a "Strategy for Infant and Young Child Feeding (IYCF) practices" to improve breastfeeding practices and ensure enforcement of national legislation related to IYCF practices [33], aligning closely with the WHO and UNICEF recommendations [1]. The Ministry of Health and Population aims to enforce the Breast Milk Substitute (Control of Sales and Distribution) Act with the help of health workers, medical students, and other cadres of the health workforce [34]. However, this act is currently under review and has been submitted to the Ministry of Law, Justice, and Parliamentary Affairs for approval pending tabling in parliament. Enforcement of this law has been poor and inconsistent, even after three decades since its endorsement [35]. Formula milk is distributed nationwide through pharmacies, grocery stores, and department stores. Such outlets and sales in urban areas are common and often unregulated due to inadequate enforcement of the existing laws [23,35]. Despite the widespread availability of formula milk, empirical research on its actual use in Nepal remains limited.

## Data source

This study used data from the NDHS for the years 2006, 2011, 2016, and 2022, which were collected by the New Era under the aegis of the Ministry of Health and Population of Nepal with technical assistance from ICF International, US. The data was retrieved from http://www.dhsprogram.com/data/available-datasets.cfm. The dataset used for this study was the individual recode for children (KR) for 2006, 2011, 2016, and 2022 [27,36–38]. The NDHS provides detailed information on breastfeeding and complementary feeding practices, as well as maternal and child health socioeconomic characteristics, collected from a nationally representative sample of households. A stratified two-stage sampling technique was employed to select the sample, and standardized validated face-to-face questionnaires were used to collect information from women of childbearing age (15–49 years). Fig 1 shows the number of participants and response rates in each survey. The total sample size analysed in this study was 7,822 mother-child pairs (0–23 months), with an average response rate of 97% (Fig 1). Details about the sampling strategies used for obtaining the information have been published elsewhere [27,36–38].

## Variables

**Outcome variable (s).** The outcome variable for this study was CMF feeding practices, defined as the proportion of children aged 0–23 months who were fed formula milk in the 24 hours preceding the interview. The prevalence of CMF

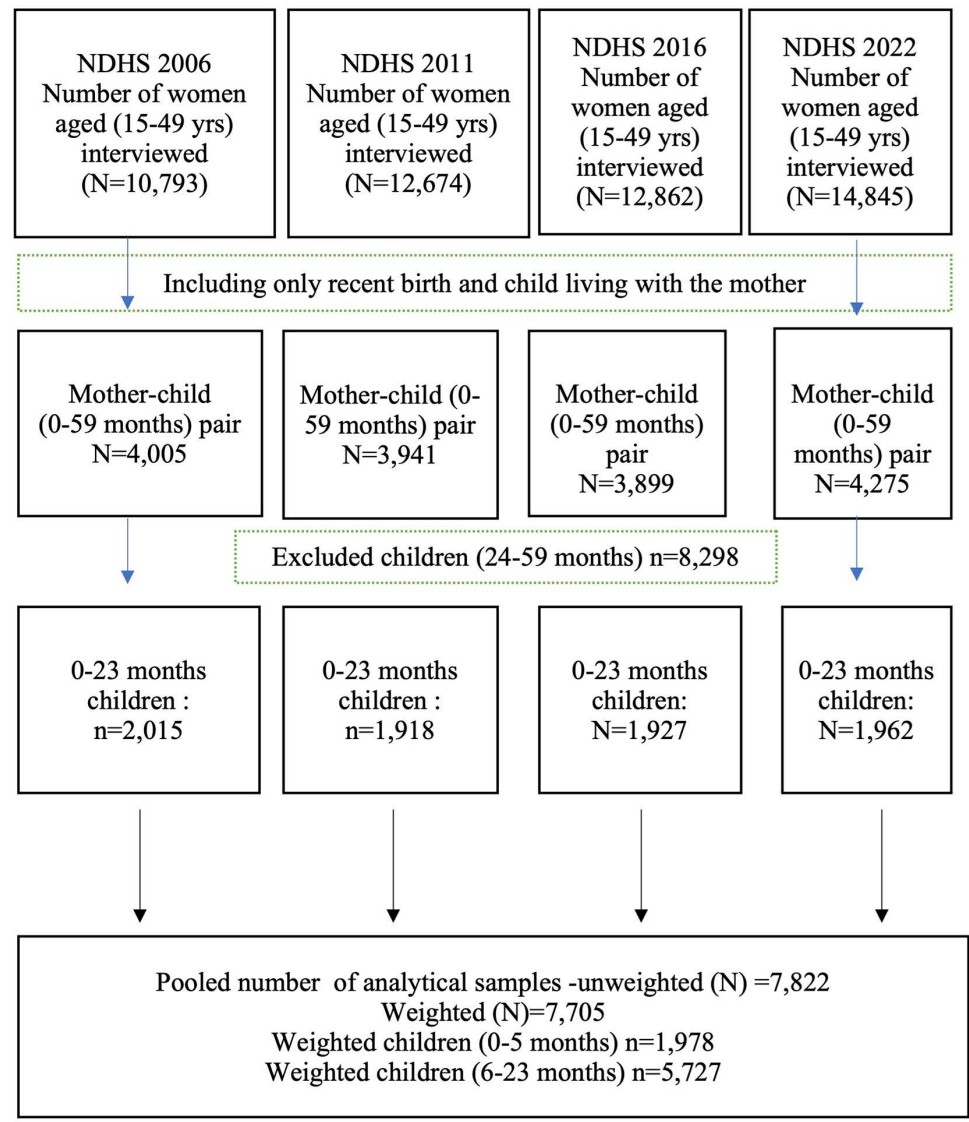

**Fig 1. Flow chart of sample size selection of study participants.**

use was further stratified into two age categories (0–5 months and 6–23 months), considering its relevance for exclusive breastfeeding and complementary feeding, as per WHO IYCF guidelines [1]. In NDHS, mothers of children aged 0–23 months were asked in the Women's Questionnaire whether their child had consumed infant formula such as Lactogen, Farex, or Nan in the previous 24 hours, alongside items like breastmilk, animal milk, and other liquids. Mothers who reported feeding their children infant formula during the 24 hours preceding the survey were classified as practicing CMF feeding ("Yes = 1"). Those who didn't were classified as not practicing formula feeding ("No = 0").

**Explanatory variables.** We chose the explanatory variables based on findings from previous South Asian studies [23,39,40] and information available in the pooled NDHS database [27,36–38]. Fig 2 illustrates the adaptation of the ecological model and the categorisation of various variables [41]. The ecological model is a theoretical framework used to understand the multiple levels of influence on individual behaviours, including interpersonal, organisational,

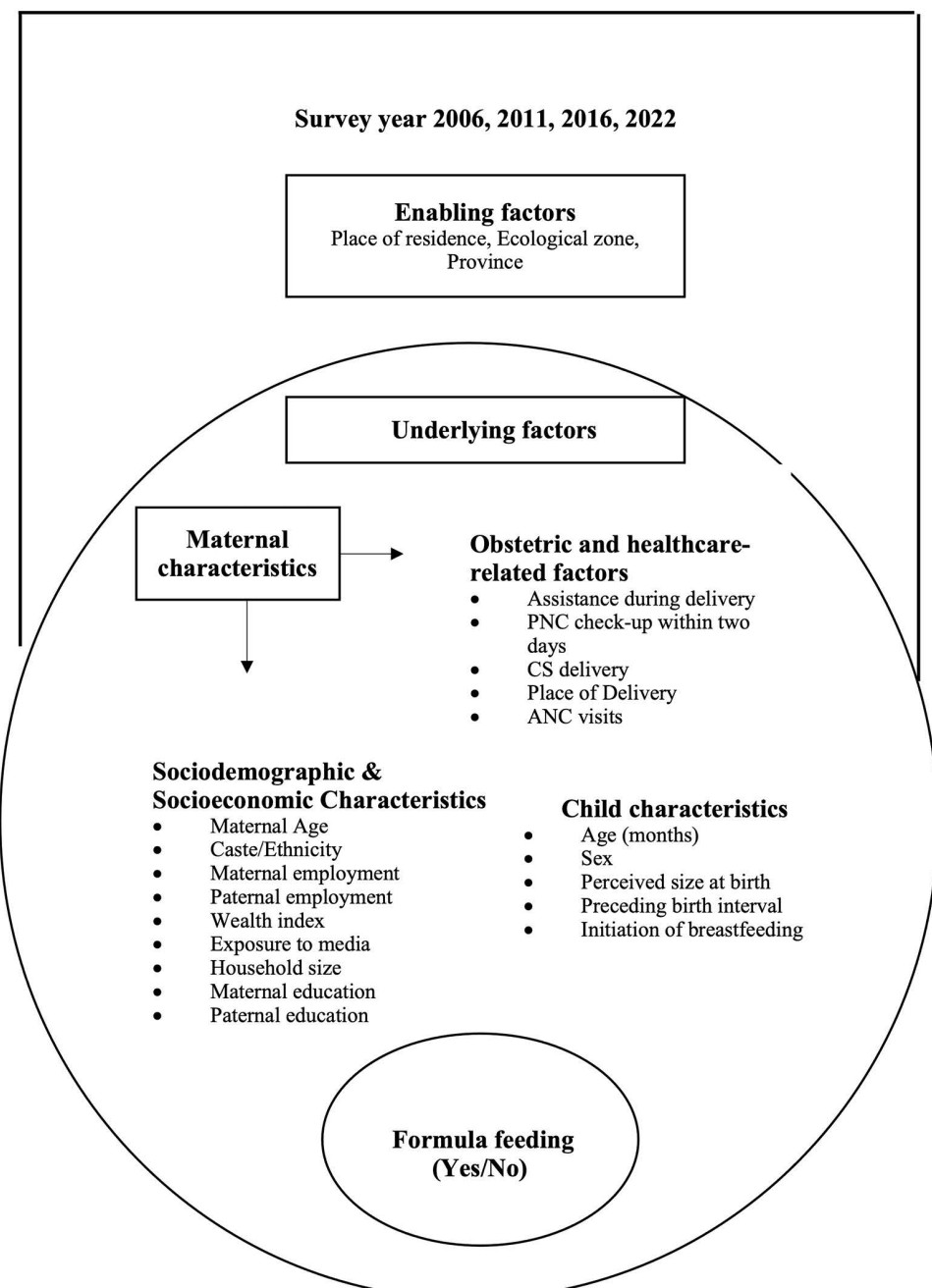

**Fig 2. Ecological model of enabling and underlying factors influencing formula feeding among children.**

community, and societal factors. It emphasises that health behaviours are shaped not only by individual-level determinants but also by the social and environmental contexts in which individuals live [42]. Addressing these interconnected layers is essential for promoting optimal feeding behaviours. This analysis included data from the four consecutive NDHS waves (2006, 2011, 2016, and 2022), with the survey year treated as a potential confounder. Variables were grouped into two broad categories: [1] enabling factors, including province (Koshi, Madhesh, Bagmati,

Gandaki, Lumbini, Karnali, Sudurpaschim), place of residence (urban/rural), and ecological zone (mountain/hill/terai); and [2] underlying factors, including maternal characteristics (age, education, employment), paternal characteristics (education, employment), socioeconomic characteristics (caste/ethnicity, wealth index, household size, exposure to media), health service related characteristics (assistance during delivery, postnatal care within two days, caesarean section birth, place of childbirth, number of ANC visits), and child characteristics (age, sex, perceived size at birth, preceding birth interval, initiation of breastfeeding).

The classification of explanatory variables is provided in Supplementary S1 Table, with certain variables described in detail for clarity. Notably, Nepal's constitution, adopted in September 2015, introduced a federal administrative structure with three tiers of government (one federal, seven provinces, and 753 local governments) to decentralize power and resources [43]. For the earlier NDHS waves (2006, 2011), the Global Positioning System (GPS) coordinates of clusters were used to identify districts, which were then grouped to align with the newly formed provinces [44,45]. Household wealth status was determined using a wealth index in all four surveys. This index was developed through principal component analysis by collecting 20–40 household asset possessions, which varied slightly across survey rounds. These include ownership of durable goods (e.g., television, refrigerator, mobile phone), housing materials (e.g., type of floor, roof, and walls), access to improved water and sanitation facilities, source of cooking fuel, and ownership of transportation means (e.g., bicycle, motorcycle, car). This method has been used across all DHS surveys conducted in Nepal and other countries [46]. In addition, mothers' education and fathers' occupations were included as key determinants of CMF feeding, based on theoretical and empirical evidence. Mother's education influences the knowledge, attitudes, and decision-making practices related to child nutrition. Higher levels of education have been associated with both increased awareness of breastfeeding benefits and, paradoxically, a greater likelihood of CMF use, particularly among mothers from urban and wealthier communities [6,47]. Likewise, a father's occupation may determine the family's socioeconomic status, time availability, and support for breastfeeding practices. Furthermore, the father's crucial role in financial decision-making can shape the accessibility and affordability of CMF, especially in contexts where fathers have higher control over the household resources [48].

## Statistical analyses

The prevalence of CMF feeding practices, along with their 95% confidence intervals for each survey year and by different age strata (0–5 and 6–23 months), was summarized using proportions and presented graphically. The distribution of formula feeding by participants' background characteristics was presented as proportions, and respective chi-square tests $x^2$) were used to assess univariable associations. Multivariable logistic regression modeling was used to identify the factors independently associated with formula-feeding practices. A two-stage technique based on a conceptual framework described by Victoria et al. [49] was employed. This technique sequentially introduced enabling and then underlying factors into the regression models to assess their independent and combined influences on formula-feeding practices. Sensitivity analysis was conducted by performing separate regression analyses for each year to assess the consistency of findings over time. All statistical analyses were performed using STATA 18 (Stata Corporation, College Station, Texas, USA) with a survey (SVY) command accounting for sample weights, primary sampling units, and strata in the complex survey design.

## Ethical considerations

This study used publicly available, de-identified secondary data from four rounds of nationally representative household surveys (NDHS 2006, 2011, 2016, and 2022). The data sets were obtained through an online application process on the DHS webpage, and permission to download and use the data was granted by MEASURE DHS/ICF International. All four surveys received ethical approval from both the Nepal Health Research Council in Nepal and the Institutional Review Board of ICF Macro in Maryland, USA.

 

## Results

### Characteristics of study participants

Table 1 presents the background characteristics of study participants, based on pooled data from the last four NDHS surveys (2006–2022). Most mother-child pairs (0–5 months and 6–23 months) were from rural settings (64.3%) and the Terai region (58.2%). Most of the children had a preceding birth interval of ≥ 24 months (50.8%), and around one out of ten (10.1%) were born through caesarean section. Regarding household and maternal characteristics, most mothers were ≤ 24 years old (56.5%), working women (58.6%), had media exposure at least once a week (46.9%), and had attained secondary education or higher (42.8%).

Fig 3 presents the unadjusted trends of CMF feeding practices from 2006 to 2022. The prevalence of CMF feeding among infants aged 0–5 months remained relatively stable between 2006 and 2011(approximately 1–2%), but showed a significant increase in 2016 and 2022, reaching 4.52% (95% CI: 2.55, 7.87) and 11.09% (95% CI: 8.01, 15.16), respectively (Fig 3A). For children aged 6–23 months, the prevalence of formula feeding gradually increased from 2.03% (95% CI: 1.26, 3.27) in 2006 to 2.54% (95% CI: 1.77, 3.65) in 2016, followed by a significant rise to 6.68% (95% CI: 5.26, 8.45) in 2022 (Fig 3B).

Figs 4 **and** 5 illustrate the unadjusted trends in CMF feeding practices among infants (0–5 months) and young children (6–23 months) across various provinces, household wealth quintiles, place of residence, caesarean deliveries, and maternal and paternal education levels between 2006 and 2022. Among all provinces, Bagmati Province observed a marked increase in CMF feeding practices among infants (0–5 m) (Fig 4A) and young children (6–23 m) (Fig 5A).

The proportion of infants (0–5 months) fed with CMF in the richest wealth quintile rose sharply from 11.81% in 2006 to 34.62% in 2022 (Fig 4B), while the proportion of children (6–23 months) in the same wealth quintile increased from 6.53% to 21.34% during the same period (Fig 5B). A similar upward trend was observed among children whose parents had attained secondary or higher education. For infants (0–5 months), the prevalence of CMF feeding among mothers with secondary and higher education increased from 6.88% in 2006 to 19.72% in 2022 (Fig 4E), while the prevalence among fathers with secondary or higher education surged from 13.04% to 35.03% (Fig 4F).

### Determinants of commercial milk formula feeding practices

Table 2 presents the distribution of CMF feeding by background characteristics. The factors significant in the Chi-squared test at $p < 0.005$ were further were further analysed using logistic regression. Table 3 presents the factors associated with formula feeding practices among children, stratified by age categories (0–5 months and 6–23 months) based on pooled data from 2006 to 2022. The full model, including non-significant variables, is provided in Supplementary S2 Table.

Among infants aged **0–5 months**, those from the NDHS 2022 were 5.40 times more likely to be fed formula milk (adjusted odds ratio (aOR): 5.40; 95% confidence interval (CI): 1.41, 0.77) compared to those from the 2006 survey. Infants born to mothers with secondary or higher education (aOR: 12.48; 95% CI: 2.63, 59.14), mothers aged 25–34 years (aOR: 2.29; 95% CI: 1.14, 4.60), and those delivered by caesarean section (aOR: 2.16; 95% CI: 1.01, 4.59) were more likely to receive formula milk compared to their counterpart. In contrast, infants born to households with middle wealth quintile (aOR: 0.17; 95% CI: 0.03, 0.84) relative to poorest wealth quintile and belonging to Karnali (aOR: 0.04; 95% CI: 0.01, 0.39) and Sudurpaschim (aOR: 0.06; 95% CI: 0.01, 0.33) provinces were less likely to receive formula milk compared to those in Koshi province (Table 3).

Among children aged 6–23 months, six factors were significantly associated with the use of CMF. Multivariable analysis indicated that those children who were male (aOR: 1.67; 95% CI: 1.15, 2.42), perceived as small size at birth (aOR: 2.76; 95% CI: 1.48, 5.15), first born (aOR: 2.59; 95% CI: 1.02, 6.56), born in the health facility (aOR: 3.18; 95% CI: 1.16, 8.73), and whose fathers had secondary or higher education (aOR: 3.14; 95% CI: 1.22, 8.10) were more likely to receive formula feed compared to their counterparts. Children born to poorer (aOR: 3.08; 95% CI: 1.35, 7.01), middle (aOR: 3.02; 95% CI:

**Table 1. Background characteristics of mother-child pair stratified by age – pooled data analysis 2006-2022) Weighted N = 7,705).**

| Variables | Child age (months) | |
| --- | --- | --- |
| | 0-5 | 6-23 |
| | Weighted frequency (%) | Weighted frequency (%) |
| **N (weighted)** | **1,978** | **5,727** |
| **Formula feeding practice (Yes)** | **94 (4.7)** | **191 (3.3)** |
| **Survey year** | | |
| 2006 | 478 (24.2) | 1,428 (24.9) |
| 2011 | 530 (26.8) | 1,439 (25.1) |
| 2016 | 443 (22.4) | 1,495 (26.1) |
| 2022 | 527 (26.6) | 1,365 (23.8) |
| **Enabling factors** | | |
| **Place of residence** | | |
| Urban | 705 (35.7) | 1,997 (34.9) |
| Rural | 1,272 (64.3) | 3,730 (65.1) |
| **Ecological zone** | | |
| Mountain | 128 (6.5) | 452 (7.9) |
| Hill | 699 (35.3) | 2,208 (38.6) |
| Terai | 1,151 (58.2) | 3,067 (53.6) |
| **Province** | | |
| Koshi | 354 (17.9) | 1,057 (18.5) |
| Madhesh | 454 (23.0) | 1,191 (20.8) |
| Bagmati | 327 (16.6) | 987 (17.2) |
| Gandaki | 191 (9.6) | 525 (9.2) |
| Lumbini | 298 (15.1) | 957 (16.7) |
| Karnali | 116 (5.9) | 410 (7.2) |
| Sudurpaschim | 237 (12.0) | 599 (10.5) |
| **Underlying factors** | | |
| **Infant characteristics** | | |
| **Child sex** | | |
| Male | 1,037 (52.4) | 2,992 (52.2) |
| Female | 941 (47.6) | 2,735 (47.8) |
| **Perceived baby size** | | |
| Small | 334 (17.0) | 941 (16.4) |
| Average | 1,352 (68.6) | 3,816 (66.7) |
| Large | 284 (14.4) | 967 (16.9) |
| **Preceding birth interval** | | |
| No previous birth | 741 (37.5) | 2,136 (37.3) |
| <24 months | 231 (11.7) | 718 (12.5) |
| >=24 months | 1,005 (50.8) | 2,873 (50.2) |
| **Initiation of breastfeeding** | | |
| More than an hour | 1,048 (53.0) | 2,964 (51.7) |
| Immediately | 930 (47.0) | 2,763 (48.3) |
| **Obstetric and health service related characteristics** | | |
| **Assistance during childbirth** | | |
| Health personnel | 967 (49.3) | 2,683 (47.3) |

*(Continued)*

**Table 1.** (Continued)

| Variables | Child age (months) | |
| --- | --- | --- |
| | **0-5** | **6-23** |
| | **Weighted frequency (%)** | **Weighted frequency (%)** |
| TBA/Relative/Others | 915 (46.7) | 2,701 (47.7) |
| No One | 78 (4.0) | 284 (5.0) |
| **PNC check within two days for mother** | | |
| No | 1,179 (59.6) | 3,459 (60.4) |
| Yes | 799 (40.4) | 2,268 (39.6) |
| **Caesarean section birth** | | |
| No | 1,778 (89.9) | 5,203 (90.8) |
| Yes | 199 (10.1) | 524 (9.2) |
| **Place of child birth** | | |
| Elsewhere* | 913 (46.2) | 2,788 (48.7) |
| Health facilities | 1,064 (53.8) | 2,939 (51.3) |
| **Antenatal visits** | | |
| < 4 ANC visits | 792 (40.0) | 2,345 (40.9) |
| >= 4 ANC visits | 1186 (60.0) | 3,382 (59.1) |
| **Sociodemographic and household characteristics** | | |
| **Maternal age (years)** | | |
| <24 | 1,118 (56.5) | 2,916 (50.9) |
| 25-34 | 758 (38.3) | 2,393 (41.8) |
| 35-49 | 101 (5.1) | 418 (7.3) |
| **Caste/ethnicity** | | |
| Brahmin/Chhetri | 555 (28.1) | 1,590 (27.8) |
| Madheshi | 323 (16.4) | 950 (16.6) |
| Dalit | 338 (17.1) | 876 (15.3) |
| Janajati | 617 (31.2) | 1,949 (34.0) |
| Muslim | 144 (7.3) | 361 (6.3) |
| **Maternal employment status** | | |
| Currently not working | 819 (41.4) | 1,995 (34.8) |
| Currently Working | 1,159 (58.6) | 3,731 (65.2) |
| **Paternal employment status** | | |
| Currently not working | 58 (2.9) | 157 (2.7) |
| Currently Working | 1,918 (97.1) | 5,549 (97.3) |
| **Wealth index** | | |
| Poorest | 436 (22.0) | 1,343 (23.5) |
| Poorer | 403 (20.4) | 1,224 (21.4) |
| Middle | 449 (22.7) | 1,225 (21.4) |
| Richer | 397 (20.1) | 1,093 (19.1) |
| Richest | 292 (14.8) | 842 (14.7) |
| **Media exposure** | | |
| Not at all | 426 (21.6) | 1,138 (19.9) |
| Less than once a week | 620 (31.4) | 1,881 (32.9) |
| At least once a week | 925 (46.9) | 2,698 (47.2) |

*(Continued)*

**Table 1.** (Continued)

| Variables | Child age (months) | |
|---|---|---|
| | 0-5 | 6-23 |
| | Weighted frequency (%) | Weighted frequency (%) |
| **Household size** | | |
| 1-3 | 140 (7.1) | 656 (11.4) |
| 4-5 | 627 (31.7) | 1,911 (33.4) |
| 6-38 | 1,210 (61.2) | 3,161 (55.2) |
| **Maternal education** | | |
| No education | 653 (33.0) | 2,118 (37.0) |
| Primary | 478 (24.2) | 1,302 (22.7) |
| Secondary and higher | 846 (42.8) | 2,307 (40.3) |
| **Paternal education** | | |
| No education | 345 (17.4) | 984 (17.2) |
| Primary education | 1,396 (70.6) | 4,051 (71.0) |
| Secondary and higher | 237 (12.0) | 673 (11.8) |

1.40, 6.51), richer (aOR: 3.01; 95% CI: 1.31, 6.92) and richest wealth quintile (aOR: 5.87; 95% CI: 2.42, 14.22) households were more likely to be formula-fed compared to those from the poorest wealth quintile (Table 3).

## Discussion

The primary objective of this study was to assess the trends and determinants of CMF feeding practices using the nationally representative NDHS data from 2006 to 2022. Despite the significant implications of CMF on maternal and child health, it is not regularly monitored as an indicator in Nepal. To our knowledge, this is the first comprehensive study to examine CMF feeding trends and determinants in the country.

Our analysis shows a substantial increase in CMF feeding among infants aged 0–5 months in Nepal between 2006 and 2022. Considering the consequences of CMF on human and planetary health [12,16,17], this trend is concerning, as it indicates a shift away from the recommended EBF practice. This pattern aligns with global increases in CMF use, with global sales approaching US$55 billion annually by 2023 [2]. Our findings align with recent studies reporting a decline in EBF rates in Nepal from 2011 to 2022, as noted in the NDHS [27,37,38]. Similar increases in CMF use and challenges to breastfeeding practices have been documented in other South Asian countries, including India [50] and Bangladesh [51], underscoring the shared regional challenges in promoting optimal infant feeding practices.

Despite multiple government initiatives, including the National Nutrition Strategy (2004), the 2014 IYCF strategy, and the Multi-Sector Nutrition Plan-III (2023–2030), Nepal continues to face significant challenges [52]. These findings have direct implications for achieving Nepal's commitment to SDG and the WHO Nutrition Targets, which emphasize optimal breastfeeding practices as a cornerstone of child survival and development. Strengthening enforcement of CMF marketing regulations and expanding breastfeeding support programs are critical to reducing malnutrition and improving child health outcomes.

The increasing use of infant formula reflects shifting societal norms, urbanization, and economic development, which impose a substantial financial burden on households. During our study, the cheapest formula milk available on the market ranged from 800 to 850 Nepali rupees (~$6.0 to $ 6.5). Given that approximately 20% of Nepal's population lives below the poverty line [53], CMF remains unaffordable for most households. The high cost of CMF relative to household income and poverty level exacerbates health inequities and undermines poverty reduction efforts, making breastfeeding promotion a key public health priority.

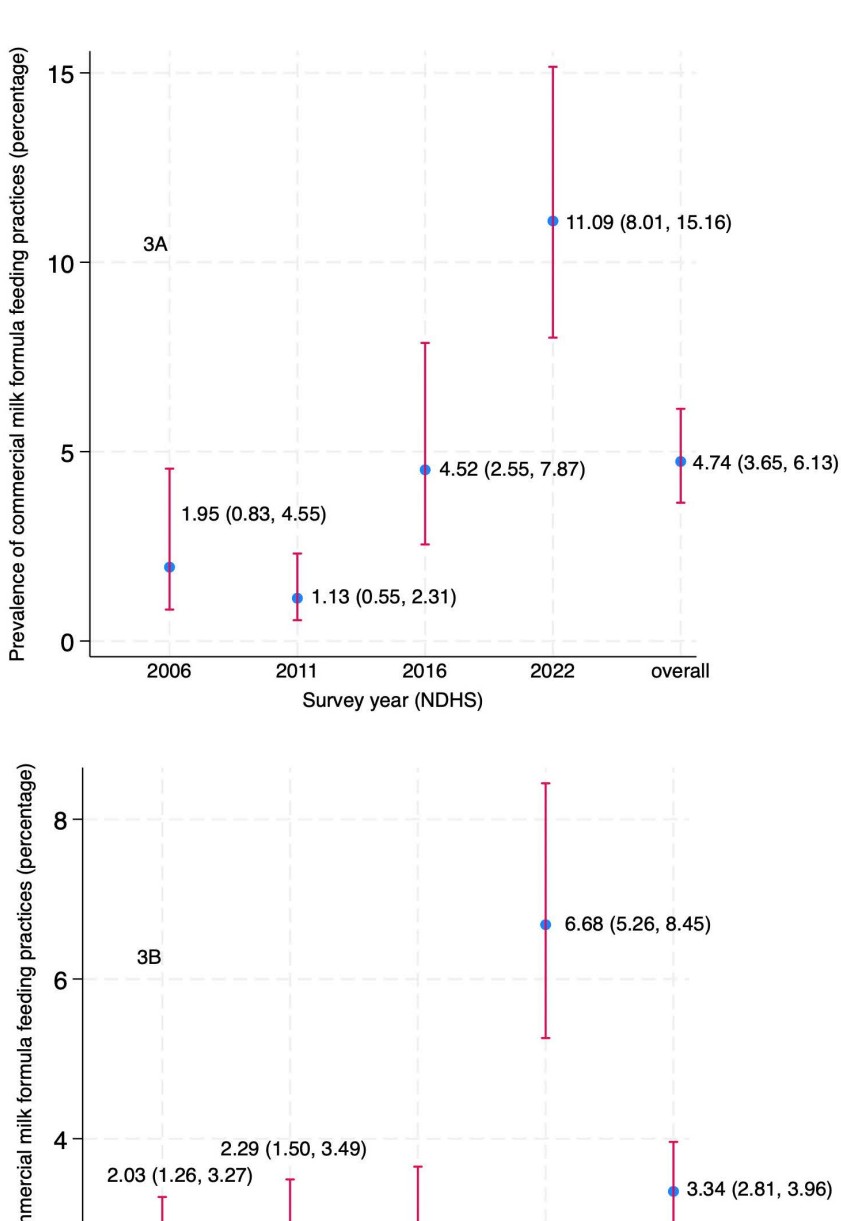

**Fig 3. Trends in commercial milk formula feeding practices among children (2006–2022).**

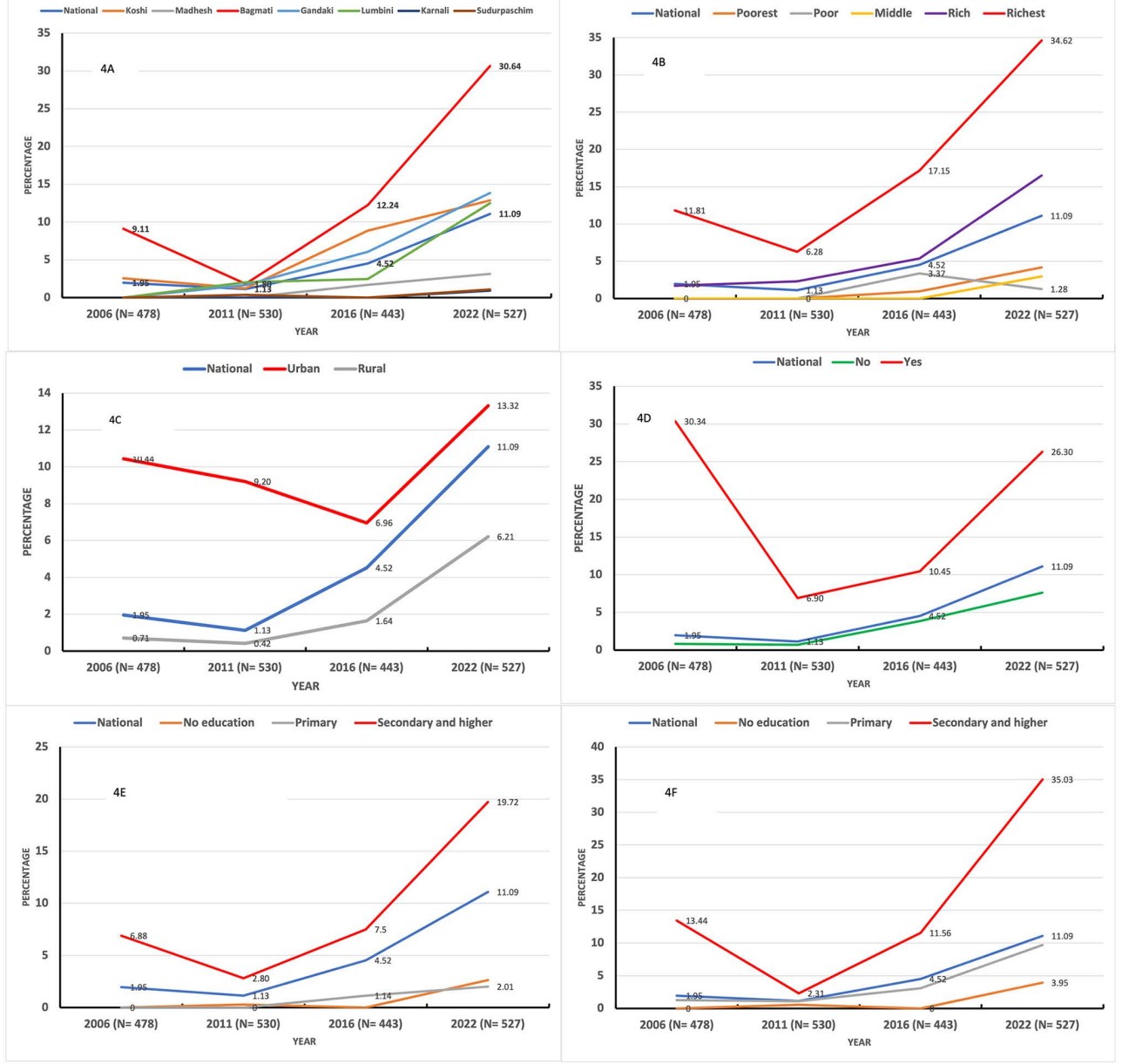

**Fig 4. Trends in commercial milk formula feeding practices among 0-5 m infants by background characteristics (2006–2022).**

Our study identified several factors significantly associated with CMF use among children aged 0–5 months and 6–23 months. Province of residence, parental education, child sex, size at birth, birth interval, place of childbirth, maternal age, and household wealth index were key determinants, highlighting the complex multifactorial influences of demographic, socio-economic, and healthcare-related practices in Nepal.

Significant provincial disparities in CMF feeding were observed. Infants aged 0–5 months in Karnali and Sudurpaschim provinces are less likely to be CMF-fed than those in Koshi province. These provinces have the lowest urbanization nationally [54], aligning with the evidence that urbanization is strongly correlated with CMF, both within Nepal [23] and

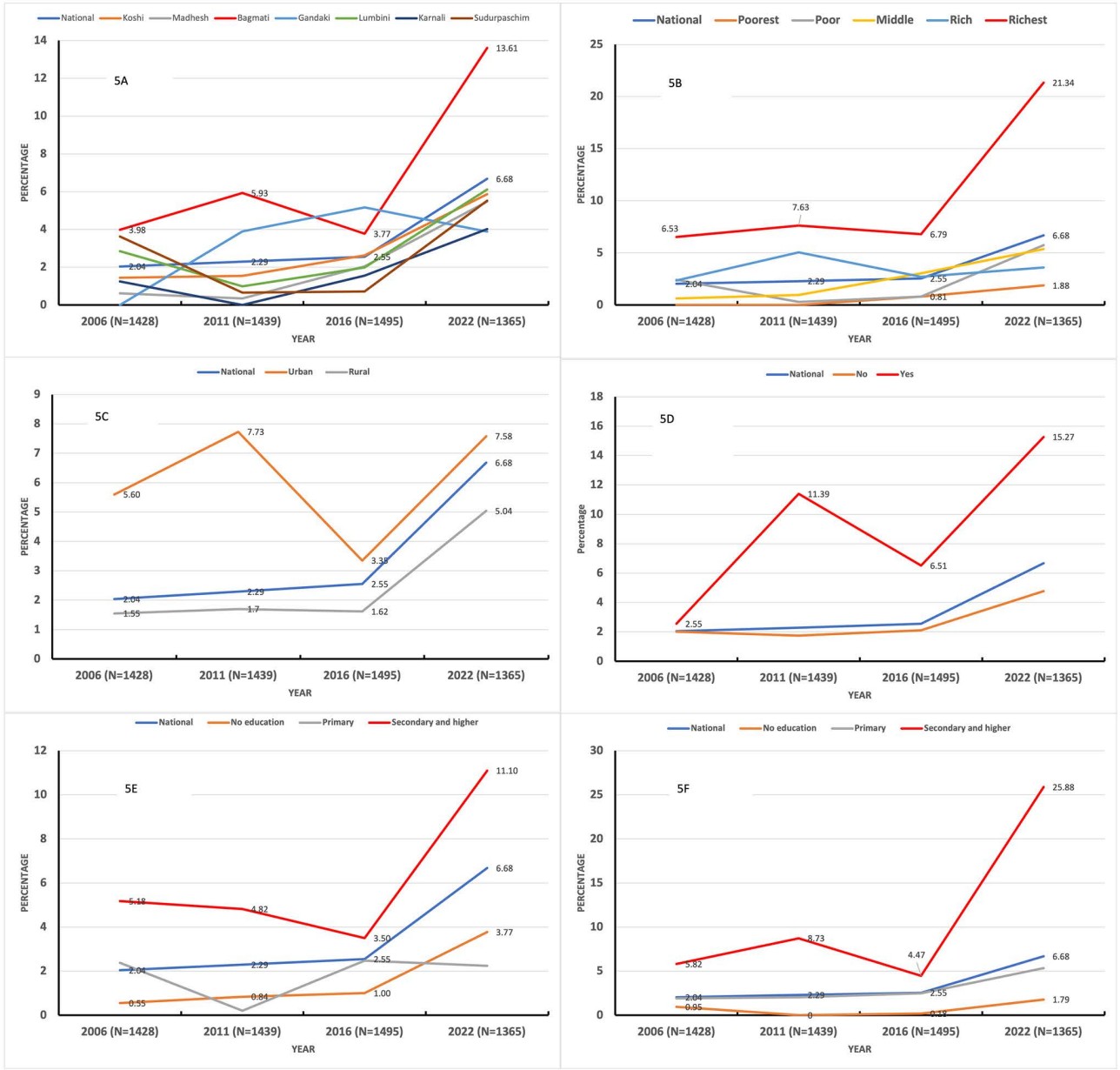

**Fig 5. Trends in commercial milk formula feeding practices among 6-23 m children by background characteristics (2006–2022).**

globally [12,13,22]. Specifically, the higher prevalence of CMF in more urbanized areas of Nepal, such as Koshi Province, is consistent with evidence from other regions. Baker et al. and Robes et al [12,13] highlight how the proliferation of corporate marketing of CMF is particularly effective in changing feeding environments in urban settings within low- and middle-income countries. Urban areas often provide increased physical proximity to CMF products, greater exposure to aggressive marketing, and may offer a perceived effective solution to modern day real world challenges, such as higher rates of maternal work outside of the home, shrinking family sizes (shift from larger kin-dominated to smaller nuclear units), and the adoption of 'modernity' values and lifestyles more conducive to formula-feeding.

**Table 2. Proportion of commercial milk formula feeding practices by background characteristics, NDHS pooled data 2006-2022 (weighted N = 7,705).**

| Variables | 0-5 months (N = 1,978) | 6-23 months (N = 5,727) |
|---|---|---|
| | *n (%)* | *n (%)* |
| **Survey year** | **p < 0.001** | **p < 0.001** |
| NDHS 2006 | 9 (1.9) | 29 (2.0) |
| NDHS 2011 | 6 (1.1) | 33 (2.3) |
| NDHS 2016 | 20 (4.5) | 38 (2.5) |
| NDHS 2022 | 58 (11.1) | 91 (6.7) |
| **Enabling factors** | | |
| **Place of residence** | **p < 0.001** | **p < 0.001** |
| Urban | 75 (10.7) | 114 (5.7) |
| Rural | 19 (1.5) | 77 (2.1) |
| **Ecological zone** | p = 0.471 | p = 0.073 |
| Mountain | 3 (2.0) | 8 (1.7) |
| Hill | 39 (5.6) | 88 (4.0) |
| Terai | 52 (4.5) | 95 (3.1) |
| **Province** | **p < 0.001** | **p < 0.001** |
| Koshi | 23 (6.5) | 29 (2.8) |
| Madhesh | 6 (1.4) | 29 (2.5) |
| Bagmati | 42 (12.9) | 64 (6.6) |
| Gandaki | 8 (4.4) | 16 (3.0) |
| Lumbini | 13 (4.2) | 29 (3.0) |
| Karnali | 0.3 (0.3) | 7 (1.7) |
| Sudurpaschim | 1 (0.4) | 17 (2.8) |
| **Underlying factors** | | |
| **Infant characteristics** | | |
| **Child sex** | p = 0.946 | **p = 0.002** |
| Male | 49 (4.7) | 125 (4.2) |
| Female | 45 (4.8) | 66 (2.4) |
| **Perceived size at birth** | p = 0.892 | p = 0.057 |
| Small | 14 (4.2) | 44 (4.7) |
| Average | 68 (5.0) | 124 (3.3) |
| Large | 12 (4.4) | 23 (2.4) |
| **Preceding birth interval** | **p < 0.01** | **p < 0.001** |
| No previous birth | 47 (6.3) | 99 (4.6) |
| <24 months | 1 (0.6) | 5 (0.8) |
| >=24 months | 45 (4.5) | 87 (3.0) |
| **Initiation of breastfeeding** | p = 0.147 | p = 0.877 |
| more than 1 hour | 59 (5.6) | 100 (3.4) |
| immediately | 35 (3.8) | 91 (3.3) |
| **Obstetric and health service related characteristics** | | |
| **Assistance during delivery** | **p < 0.001** | **p < 0.001** |
| Health personnel | 80 (8.3) | 149 (5.6) |
| TBA/Relative/Others | 13 (1.4) | 41 (1.5) |
| No One | 1 (1.6) | 1 (0.5) |
| **PNC check within two days** | **p < 0.001** | **p < 0.001** |
| No | 20 (1.7) | 60 (1.8) |

*(Continued)*

**Table 2.** (Continued)

| Variables | 0-5 months (N = 1,978) | 6-23 months (N = 5,727) |
|---|---|---|
| | *n (%)* | *n (%)* |
| Yes | 74 (9.2) | 131 (5.8) |
| **Delivery by caesarean section** | **p < 0.001** | **p < 0.001** |
| No | 55 (3.1) | 133 (2.6) |
| Yes | 39 (19.4) | 58 (11.1) |
| **Place of child birth** | **p < 0.001** | **p < 0.001** |
| Elsewhere | 55 (3.1) | 24 (0.9) |
| health facilities | 39 (19.4) | 167 (5.7) |
| **Antenatal visits** | **p < 0.001** | **p < 0.001** |
| Less than 4 ANC | 13 (1.6) | 28 (1.2) |
| >=4 ANC | 81 (6.8) | 163 (4.8) |
| **Sociodemographic and household characteristics** | | |
| **Maternal age (years)** | **p < 0.001** | **p < 0.001** |
| <24 | 27 (2.4) | 78 (2.7) |
| 25-34 | 62 (8.2) | 104 (4.3) |
| 35-49 | 4 (4.4) | 9 (2.2) |
| **Caste/Ethnicity** | **p = 0.026** | **p < 0.001** |
| Brahmin/Chhetri | 39 (7.0) | 81 (5.0) |
| Madheshi | 8 (2.5) | 28 (2.9) |
| Dalit | 7 (2.0) | 13 (1.4) |
| Janajati | 35 (5.6) | 63 (3.2) |
| Muslim | 6 (4.1) | 6 (1.8) |
| **Maternal employment status** | p = 0.301 | **p = 0.03** |
| Currently not working | 45 (5.5) | 84 (4.2) |
| Currently Working | 49 (4.2) | 107 (2.9) |
| **Paternal employment status** | p = 0.262 | p = 0.939 |
| Currently not working | 5 (7.9) | 5 (3.5) |
| Currently Working | 88 (4.6) | 186 (3.3) |
| **Wealth index** | **p < 0.001** | **p < 0.001** |
| Poorest | 5 (1.2) | 8 (0.6) |
| Poorer | 4 (1.1) | 28 (2.3) |
| Middle | 3 (0.7) | 30 (2.4) |
| Richer | 26 (6.6) | 37 (3.4) |
| Richest | 55 (18.8) | 88 (10.5) |
| **Media exposure** | **p = 0.029** | **p < 0.001** |
| Not at all | 11 (2.5) | 23 (2.0) |
| Less than once a week | 23 (3.7) | 42 (2.2) |
| At least once a week | 57 (6.2) | 126 (4.7) |
| **Household size** | **p = 0.042** | p = 0.262 |
| 1-3 | 15 (10.4) | 14 (2.2) |
| 4-5 | 31 (5.0) | 63 (3.3) |
| 6-38 | 47 (3.9) | 114 (3.6) |
| **Maternal education** | **p < 0.001** | **p < 0.001** |
| No education | 3 (0.4) | 24 (1.1) |
| Primary | 5 (1.0) | 24 (1.9) |
| secondary and higher | 86 (10.2) | 143 (6.2) |

*(Continued)*

**Table 2.** (Continued)

| Variables | 0-5 months (N = 1,978) | 6-23 months (N = 5,727) |
|---|---|---|
| | n (%) | n (%) |
| **Paternal education** | **p < 0.001** | **p < 0.001** |
| No education | 3 (0.8) | 24 (0.6) |
| Primary | 59 (4.2) | 24 (3.0) |
| Secondary and higher | 32 (13.5) | 143 (9.5) |

These findings underscore the urgent need for robust policy enforcement regarding the marketing and distribution of CMF. Nepal adopted the Breastmilk Substitutes (Marketing Control) Act 2049 in 1992 and Regulation 2051 in 1994 to clarify provisions, including restrictions in the health system, required procedures for labelling approval, and monitoring and inspection [55]. However, weak enforcement has enabled the widespread availability of CMF through pharmacies and grocery stores, particularly in urban areas. Policymakers should strengthen regulatory bodies, impose stricter penalties for violations, and monitor online and indirect marketing tactics that circumvent traditional advertising bans.

Higher parental education was associated with increased CMF use, reflecting the complex interaction between education, employment, and childcare practices. Increased female education in LMICs often correlates with higher labour force participation [56]. Returning to work is linked to poorer breastfeeding practices in the absence of a supportive workplace environment [57]. One crucial factor contributing to this, particularly for working mothers, is the duration and accessibility of maternity leave. In Nepal, maternity leave is officially three months, primarily for women with permanent government positions, while the private and unorganized sectors often provide no such benefits. This limited leave frequently compels mothers to introduce CMF, creating a significant tension between professional and carer responsibilities that impedes optimal breastfeeding. Therefore, a critical policy implication is the urgent need to universalize and enforce comprehensive paid maternity leave across all employment and to promote breastfeeding-friendly workplaces with accessible lactation facilities and flexible work arrangements. Similar trends have been documented globally, with studies reporting a significant negative association between mothers' education and EBF practices in Pakistan, China, Saudi Arabia, the United Arab Emirates, and Lebanon [58–62].

We were unable to compare the association between a father's education and CMF feeding for children aged 6–23 months due to the unavailability of relevant literature. For infants aged 0–5 months, those from middle-income quintile households were less likely to be formula-fed than those from the poorest households. A systematic review examining the enablers and barriers to appropriate IYCF practices in India found that higher material socioeconomic status (SES) was an enabler for EBF [63]. For children aged 6–23 months, we found increased odds of formula feeding in the richest quintiles, aligning with a multi-country study that examined data from 90 LMICs and confirmed significantly higher formula use in wealthier groups [63].

There was also an association between the child's gender and birth order with infant formula feeding. Male infants were more likely to receive CMF during 6–23 months, possibly reflecting cultural preferences for male children within South Asian societies, including Nepal. Firstborn children across both age groups (0–5 months and 6–23 months) were more likely to receive CMF, which may be attributed to the mother's inexperience with breastfeeding [30]. This aligns with literature suggesting that multiparous mothers produce colostrum and breast milk earlier, facilitating EBF [64]. While our finding is consistent with studies reporting higher formula feeding among first-born children [64,65], a 2015 review on EBF determinants in developing countries reported an inconsistent association between parity and EBF [66].

Our study revealed that small size at birth is associated with a higher likelihood of CMF feeding during 6–23 months, consistent with its recognized role as a risk factor for non-exclusive breastfeeding. Although comparable literature for this

**Table 3. Association of enabling and underlying factors with infant formula feeding practices among children aged 0-5 months and 6-23 months.**

| Variables | 0-5 months | 6-23 months |
|---|---|---|
| | Adjusted odds ratio (95% CI) | Adjusted odds ratio (95% CI) |
| **Survey year** | | |
| NDHS 2006 | 1.00 | 1.00 |
| NDHS 2011 | 0.73 (0.18, 2.93) | 0.86 (0.39, 1.90) |
| NDHS 2016 | 1.50 (0.33, 6.86) | 0.60 (0.26, 1.41) |
| NDHS 2022 | 5.40 (1.41, 20.77)** | 1.68 (0.80, 3.66) |
| **Province** | | |
| Koshi | 1.00 | 1.00 |
| Madhesh | 0.35 (0.08, 1.48) | 1.20 (0.55, 2.65) |
| Bagmati | 1.50 (0.71, 3.19) | 1.75 (0.99, 3.11) |
| Gandaki | 0.63 (0.23, 1.68) | 1.11 (0.52, 2.34) |
| Lumbini | 0.68 (0.26, 1.79) | 1.04 (0.54, 2.00) |
| Karnali | 0.04 (0.01, 0.39)** | 1.10 (0.49, 2.46) |
| Sudurpaschim | 0.06 (0.01, 0.33)** | 1.34 (0.57, 3.11) |
| **Child sex** | NS | |
| Female | | 1.00 |
| Male | | 1.67 (1.15, 2.42)** |
| **Perceived size at birth** | NS | |
| Large | | 1.00 |
| Small | | 2.76 (1.48, 5.15)** |
| Average | | 1.31 (0.74, 2.32) |
| **Preceding birth interval** | NS | |
| <24 months | | 1.00 |
| No previous birth | | 2.59 (1.02, 6.56)* |
| >=24 months | | 2.51 (0.99, 6.37) |
| **Place of child birth** | NS | |
| Elsewhere | | 1.00 |
| Health facilities | | 3.18 (1.16, 8.73)* |
| **Maternal age (years)** | | |
| <24 | 1.00 | 1.00 |
| 25-34 | 2.29 (1.14, 4.60)* | 1.29 (0.89, 1.86) |
| 35-49 | 3.41 (0.58, 20.07) | 1.04 (0.43, 2.53) |
| **Wealth index** | | |
| Poorest | 1.00 | 1.00 |
| Poorer | 0.42 (0.09, 1.98) | 3.08 (1.35, 7.01)** |
| Middle | **0.17 (0.03, 0.84)*** | 3.02 (1.40, 6.51)** |
| Richer | 1.63 (0.51, 5.25) | 3.01 (1.31, 6.92)** |
| Richest | 2.13 (0.55, 8.30) | 5.87 (2.42, 14.22)*** |
| **Maternal education** | | |
| No education | 1.00 | 1.00 |
| Primary | 1.39 (0.24, 8.10) | 0.79 (0.38, 1.65) |
| Secondary and higher | 12.48 (2.63, 59.14)** | 1.50 (0.73, 3.08) |
| **Paternal education** | | |
| No education | 1.00 | 1.00 |
| Primary | 0.69 (0.11, 4.38) | 2.22 (0.93, 5.33) |

*(Continued)*

**Table 3.** (Continued)

| Variables | 0-5 months | 6-23 months |
|---|---|---|
| | **Adjusted odds ratio (95% CI)** | **Adjusted odds ratio (95% CI)** |
| Secondary and higher | 0.96 (0.14, 6.48) | 3.14 (1.22, 8.10)* |

\***Significant at p-value < 0.001. **Significant at p-value < 0.01. *Significant at p-value < 0.05; 1.00 represents the reference category. NS: not significant. **Variables included in the model for 0–5 months infants:** *survey year, place of residence, province, child sex, preceding birth interval, initiation of breast-feeding, provider of delivery during labour, PNC check within two days, delivery by C/S, place of childbirth, ANC visits, Maternal age, Caste/ethnicity, Wealth index, Media exposure, Household size, Maternal educa-tion, Paternal education;* **Variables included in the model for 6–23 months children**: *Survey year, Place of residence, Province, Child sex, Perceived size at birth, Preceding birth interval, Provider of delivery during labour, PNC check within 2 days, Delivery by C/S, Place of childbirth, Maternal age, Caste/ethnicity, Maternal employment, Wealth index, Media exposure, Maternal education, Paternal education household size, maternal education, paternal education.* Full results of the models are provided in S2 Table.

group is limited, several factors may explain this trend. Small-sized infants are at a higher risk of rapid weight loss, may have slower and weaker suckling skills, and their families often have concerns about their survival, leading to the initiation of infant formula as a perceived viable alternative [67]. These findings highlight misconceptions about breastmilk ade-quacy, underscoring the need for targeted education on the benefits of breastfeeding for all infants, regardless of size [68].

## Strengths and limitations

The strength of this study lies in the use of pooled data from four rounds of nationally representative household surveys (2006–2022), providing a large sample size and improved statistical power. Sensitivity analyses confirmed the robustness of the findings, with results remaining consistent across survey years (see Supplementary S3 and S4 Tables). Further, using nationally representative NDHS data enhances the generalisability and reliability of our findings, allowing for population-level insights into evidence-based practices and disparities. Additionally, internationally recognized methods and locally validated tools were employed, resulting in a high response rate and minimizing selection bias. Finally, the IYCF data for 2006–2022 were collected using consistent core questionnaires and CMF definitions by the same institu-tions. This consistency led to the accuracy and reliability of the estimates [27,36–38].

However, this study has limitations. Firstly, the 24-hour recall method for reporting formula-feeding practices may not accurately represent an individual's usual formula-feeding practices or changes over time, potentially underestimating actual use, as it excludes children who were fed formula outside of that timeframe. Secondly, this study could not include certain potential covariates important for predicting formula feeding practices, such as maternal knowledge and attitudes toward CMF, as well as perceived insufficient breast milk, which were considered in previous studies [69]. Including these variables could affect the study findings. Thirdly, trends presented in this study of formula feeding practices over the years were unadjusted estimates, which may partly reflect changes in underlying factors rather than true shifts in the outcome of interest. Lastly, Nepal underwent significant urbanisation between 2006 and 2022, accompanied by administrative restruc-turing after 2015, which contributed to an increase in the urban population according to recent estimates. Consequently, the reported urban figure of 35% based on pooled data may not accurately represent the current urban-rural population scenario in Nepal.

## Conclusion

The prevalence of CMF feeding practices among infants and young children (0–23 months) increased from 2.02% in 2006 to 7.91% in 2022, with the sharpest rise occurring among infants aged 0–5 months (from 1.95% to 11.09%). Key determinants included maternal education, household wealth quintile, as well as child-related factors such as sex, size

at birth, and birth interval. This increasing trend poses a threat to Nepal's progress toward the WHO breastfeeding targets, with implications for child health, equity, and environmental sustainability. These findings underscore the urgent need for evidence-informed, context-specific interventions to halt and reverse the growing trend of CMF feeding in Nepal, thereby promoting and supporting optimal breastfeeding practices. Priorities include enforcing existing legislation against aggressive CMF marketing, enhancing breastfeeding support in health facilities, expanding paid maternity leave, and implementing comprehensive policies that address societal and occupational barriers to breastfeeding. Future studies should investigate the underlying drivers of formula feeding practices, particularly through qualitative research that examines cultural norms, maternal perceptions, and barriers to exclusive breastfeeding.

## Supporting information

**S1 Table. Identification and categorisation of potential variables used in the study.** Provides details on all potential variables included in the analysis and their categorisation into enabling and underlying factors.
(DOCX)

**S2 Table. Association of enabling and underlying factors with commercial milk formula feeding practices among children aged 0–5 months and 6–23 months based on pooled data analyses 2006–2022 in Nepal, 2022: results from multivariable binary logistic regression analysis.** Presents the results of multivariable binary logistic regression analyses assessing determinants of commercial milk formula feeding.
(DOCX)

**S3 Table. Sensitivity analysis of the pooled analysis on subsets of data (NDHS 2022, 2016, 2011, & 2006) among children aged 0–5 months.** Shows robustness checks using individual survey subsets for infants aged 0–5 months.
(DOCX)

**S4 Table. Sensitivity analysis of the pooled analysis on subsets of data (NDHS 2022, 2016, 2011, & 2006) among children aged 6–23 months.** Shows robustness checks using individual survey subsets for children aged 6–23 months.
(DOCX)

## Acknowledgments

The authors would like to acknowledge the Demographic and Health Survey (DHS) program for granting access to the data of the Nepal Demographic and Health Survey NDHS) 2006, 2011, 2016, and 2022.

## Author contributions

**Conceptualization:** Barun Kumar Singh, Vishnu Khanal.

**Data curation:** Barun Kumar Singh.

**Formal analysis:** Barun Kumar Singh.

**Investigation:** Vishnu Khanal.

**Methodology:** Barun Kumar Singh, Vishnu Khanal.

**Resources:** Barun Kumar Singh, Sangita Bista, Sajjan Yogesh, Vishnu Khanal.

**Software:** Barun Kumar Singh.

**Supervision:** Vishnu Khanal.

**Validation:** Barun Kumar Singh, Sangita Bista, Sajjan Yogesh, Vishnu Khanal.

**Visualization:** Barun Kumar Singh.

**Writing – original draft:** Barun Kumar Singh, Sangita Bista.

**Writing – review & editing:** Barun Kumar Singh, Sangita Bista, Sajjan Yogesh, Vishnu Khanal.

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
