## [Decision Letter · Decision Letter 0]

22 May 2025

Dear Dr. Singh,

Thank you for submitting your manuscript to PLOS ONE. After careful consideration, we feel that it has merit but does not fully meet PLOS ONE’s publication criteria as it currently stands. Therefore, we invite you to submit a revised version of the manuscript that addresses the points raised during the review process.

**ACADEMIC EDITOR: **  Authors should address the comments of the reviewers to strengthen their manuscript. 

We look forward to receiving your revised manuscript.

Kind regards,

Frank Kyei-Arthur, Ph.D.

Academic Editor

PLOS ONE

2. Please amend either the title on the online submission form (via Edit Submission) or the title in the manuscript so that they are identical.

Additional Editor Comments (if provided):

Dear Authors,

Kindly address the comments of the reviewers to strengthen your manuscript.

Reviewers' comments:

Reviewer's Responses to Questions

**Comments to the Author**

1. Is the manuscript technically sound, and do the data support the conclusions?

Reviewer #1: Partly

Reviewer #2: Yes

2. Has the statistical analysis been performed appropriately and rigorously?

Reviewer #1: No

Reviewer #2: No

3. Have the authors made all data underlying the findings in their manuscript fully available?

Reviewer #1: Yes

Reviewer #2: Yes

4. Is the manuscript presented in an intelligible fashion and written in standard English?

Reviewer #1: No

Reviewer #2: Yes

Reviewer #1: Dear editor,

The manuscript significantly advances understanding of trends in commercial infant formula use in a low-income setting, with important implications for public health policy. While well-structured, scientifically rigorous, and highly relevant to global health priorities, its originality could be enhanced by drawing comparisons with other LMICs or proposing innovative intervention strategies.

Comments:

- To improve readability and professionalism, a thorough linguistic review is necessary.

Abstract:

- Condense repetitive phrases and focus on the most important findings (e.g. fewer mentions of "commercial formula").

Introduction:

- Briefly describe Nepal's unique socio-economic and policy context with regard to breastfeeding and formula feeding.

Method:

- Provide a brief discussion of potential bias (social desirability or recall bias) and how it was reduced.

- Some of the things that were included in the analysis (for example, how well the mother was educated, and what job the father did) could be explained more to make the idea behind the analysis clearer.

Results:

- Clearly state whether subgroup trends are crude (unadjusted) or adjusted for other covariates. If trends are unadjusted, it's important to qualify this limitation and emphasise the adjusted results from multivariable models in the discussion.

- Include explicit statistical comparisons, such as tests of trend significance over time (e.g. Cochran-Armitage or Mantel-Haenszel tests) and subgroup differences (e.g. interaction terms in regression models).

- Link the results more directly to public health priorities. For example, explain how an OR of 12.48 for maternal education translates into specific policy interventions, such as targeting educated mothers with breastfeeding awareness campaigns.

- The study does not address whether the results are robust to variations in assumptions or model specifications. For example:

• Were alternative age categorizations (e.g., 0-6 months instead of 0-5 months) tested?

• Were findings consistent across different years or subsets of data?

Include sensitivity analyses to demonstrate the robustness of the findings.

- The manuscript identifies disparities across provinces but does not delve into underlying reasons (e.g., why Bagmati province has the highest formula use or why rural areas have lower prevalence).

Hypothesise potential drivers of these differences (e.g. urbanisation, marketing exposure, cultural norms) and suggest how these findings could inform targeted interventions.

-Tables & figures: Add titles or legends to clarify abbreviations and statistical measures.

Discussion:

- Expand actionable recommendations, especially targeting maternal education and urban areas where formula feeding is prevalent.

Reviewer #2: Trends in and determinants of commercial milk formula feeding practices among

children aged 0-23 months in Nepal: Pooled Analysis of four Nepal Demographic and Health Surveys (2006 - 2022)

(PONE-D-24-48446)

Dear editor,

Thank you for inviting me to review the manuscript “Trends in and determinants of commercial milk formula feeding practices among children aged 0-23 months in Nepal: Pooled Analysis of four Nepal Demographic and Health Surveys (2006 - 2022)”. The study is informative, and it has the potential to provide valuable insights for policymakers to ensure good health for infants in Sub-Saharan Africa. However, some points need to be addressed before the study can be accepted for publication.

Comments

1. Introduction: “….However, less than half (48%) of 0-5 months infants were exclusively breastfed globally 24 hours before the surveys. Does it mean that in all the 4 surveys, 48% of infants were exclusively fed, respectively? Kindly revise the statement for clarity.

Please revise the following sentence for clarity. “Only one study has directly examined the commercial milk formula used in Nepal, limited to a specific region.”

2. Regarding the dependent variable, indicate the exact question that was asked in the NDHS datasets and the original coding.

3. Methods: as part of the context, let readers know about Nepal as a country regarding its location, population size, and other socio-demographic characteristics.

4. Indicate the benefit of using nationally representative data

5. The measurement of feeding a baby with formula 24 hours before the survey seems problematic, as it cuts off those who were not fed within that period. People may be using the formula, but may not have used it within the 24 hours. This should be captured as a limitation.

6. Give a brief description of the ecological model before describing your variables with it

7. Give some examples of the several household items used to measure wealth status. It’s also better to maintain wealth status than SES.

8. In your analysis, you stated that “…which used a two-stage technique based on a conceptual framework described by Victoria et al. (40)., What is the framework about? Please spell it out for clarity. Importantly, you have highlighted that the ecological model is guiding the study. In this case, one would expect that it drives the analysis but that is missing.

9. Furthermore, looking at the nature of the model, it’s best to perform a multilevel modelling to ascertain the factors having the most impact. It will also cater for the complex nature of the sampling technique. Is there any reason for not employing that approach?

10. It’s better to first describe the socio-demographic characteristics of the respondents before other variables. This will provide contextual understanding of other subsequent findings

11. The regression Table should be labeled 3 instead of 2 to distinguish it from the chi-square table, which is labeled 2

12. From the regression Table, it appears you performed two models, one for 0-5 and 6-23, however, this is not clear in the methodology. Although you mentioned two-stage technique, this is not clear. Kindly clarify it.

13. Please insert the figures of the variables that were not significant in the regression Table

14. “Our findings align with the recent studies reporting a decreasing trend of EBF in Nepal from 2011 to 2022 NDHS”. Relating your findings to studies that used the same or a section of the NDHS is quite problematic. It is best to discuss your findings with other Asian countries.

15. One key reason that might contribute to the increase to the use of infant formula is occupational and family/child conflict. What’s the duration of maternity leave in Nepal? Don’t you think this might partly explain it? Any policy recommendation in that direction?

16. “Our findings align with studies documenting the relationship between urbanization and commercial milk formula.” Kindly provide context.

17. Generally, the discussions lack policy implications. The author(s) need to discuss important policy implications related to the findings

**Do you want your identity to be public for this peer review?** For information about this choice, including consent withdrawal, please see our Privacy Policy

Reviewer #1: No

Reviewer #2: No

---

## [Author Response · Author response to Decision Letter 1]

27 Jul 2025

Dear Reviewer and Editor

Thanks a lot for your valuable time to provide constructive feedback on our manuscript. We have tried to address all the comments in the "response to reviewer" file attached.

regards,

Barun Kumar Singh

On behalf of co-author

---

## [Decision Letter · Decision Letter 1]

9 Oct 2025

Dear Dr. Singh,

Thank you for submitting your manuscript to PLOS ONE. After careful consideration, we feel that it has merit but does not fully meet PLOS ONE’s publication criteria as it currently stands. Therefore, we invite you to submit a revised version of the manuscript that addresses the points raised during the review process.

We look forward to receiving your revised manuscript.

Kind regards,

Frank Kyei-Arthur, Ph.D.

Academic Editor

PLOS ONE

Journal Requirements:

2, Please review your reference list to ensure that it is complete and correct. If you have cited papers that have been retracted, please include the rationale for doing so in the manuscript text, or remove these references and replace them with relevant current references. Any changes to the reference list should be mentioned in the rebuttal letter that accompanies your revised manuscript. If you need to cite a retracted article, indicate the article’s retracted status in the References list and also include a citation and full reference for the retraction notice.

Additional Editor Comments:

Reviewers' comments:

Reviewer's Responses to Questions

**Comments to the Author**

Reviewer #2: (No Response)

Reviewer #3: (No Response)

2. Is the manuscript technically sound, and do the data support the conclusions?

Reviewer #2: Yes

Reviewer #3: Partly

3. Has the statistical analysis been performed appropriately and rigorously?

Reviewer #2: Yes

Reviewer #3: No

4. Have the authors made all data underlying the findings in their manuscript fully available?

Reviewer #2: Yes

Reviewer #3: Yes

5. Is the manuscript presented in an intelligible fashion and written in standard English?

Reviewer #2: No

Reviewer #3: No

Reviewer #2: Trends in and determinants of commercial milk formula feeding practices among

children aged 0-23 months in Nepal: pooled Analysis of four Nepal Demographic and Health Surveys (2006 - 2022)

PONE-D-24-48446R1

Dear Editor,

Thank you for the opportunity to review this manuscript again. The authors have addressed most of my comments, and the manuscript is now a technically sound piece of scientific research.

Nonetheless, there are some additional comments that should be addressed.

Comments to Authors

1.The title/topic is quite long. Kindly try to shorten it.

2.“….However, less than half (48%) of 0-5 months infants were exclusively breastfed

3.globally 24 hours before the surveys. “Author response: Thank you for seeking clarity. We have presented it as a global prevalence based on UNICEF-provided figures, which they update periodically based on different survey estimates.” In this case, it would be better to state the year and mention UNICEF’s report in the statement to better understand the survey you are referring to.

4.Reference 4 has been repeated in the reference list. It appears you stated WHO but referenced UNICEF. Kindly recheck the entire references for corrections where necessary.

5.Lines 32 to 34“…..this growth is expected to continue, with sales projected to rise by another 10.8%, reaching 2.38 million tonnes by 2024 (12), reflecting a situation where more infants and young children are provided.” We are currently in 2025, hence you cannot project to 2024. Kindly revise it.

6.Lines 158 to 164 seem incomplete. Recheck and complete it for easy understanding. Besides, the word firstly should be followed by secondly or an alternative phrase

7.Line 198:” and respective chi-square tests ( were employed” recheck the sentence. There is an omission from the bracket.

8.199-202: Specify the two-stage techniques you used. “Specifically, we sequentially introduced enabling and then underlying factors into the regression models to assess their independent and combined influence on formula feeding practices. We have now revised the Statistical Analyses section to clarify this distinction. Please check Page no 9; Para 2; Line number 199-202. Your response to my previous comment gives insight into the section, but it’s not captured in the manuscript.

9.Lines 218 to 224 give some statistics in your description. Line 223, insert a comma after working women to make the sentence clearer

10.Line 231, delete in 2016 and 2022 after respectively

11.Give appropriate labeling of your figures. For example, Figure 1: Flow chart of sample size selection of participants. Besides, label each figure under the referred figure for easy identification. You have stated Figure 3, Figure 3A, and Figure 3B, but these cannot be found since they have been labeled as A and B under Figure 3. Kindly label each as indicated in the write-up.

12.Line 244: insert with after fed

13.Line 245 ….. in 2011 but significantly increased to 30.64 %...” Which year are you referring to? 2022? Also, correct the sentence beginning with “while”

14.The Tables have still not been labeled well. As indicated earlier, the regression analysis is labeled 2, likewise the cross-tab Table.

15.Line 268, revise the sentence for understanding

16.Lines 307-311 should be a continuous sentence

17.line 336, change ‘must’ to should

18.Line 432: Be specific on the vulnerable groups or target population. Also, indicate the people to enforce the stated policies or interventions. Recommend also for future studies. For example, do you perceive that a qualitative analysis on the issue will be useful?

19.There are still typographical errors in the manuscript that need correction. I have indicated a few but the authors need to work on the errors

Reviewer #3: Dear authors

This article is important for policymakers in the health sectors , I think urgent actions and steps needed to control this situation .I have some comments General comments

1. This article needs linguistic review , and formatting editing .

2. There some sentences were used many times , So it must be abbreviated

Introduction

1. Describe more information on demography and Geography of Nepal

2. Can you explain the reasons behind the out of law places of Saling the commercial milk formula milk in Nepal even in Urban areas where the government suppose to have more authorities and power. Is there a role for gangs, smugglers, foreign affairs or the law?

3. You mention that less than half (48%) of 0-5 months infants were exclusively breastfed globally 24 hours before the surveys. Does it mean that it is real exclusive BF?

4. The commercial milk formula that are present in the local markets in Nepal locally produced or imported?

Methods

1. Context of the study it is better to be shifted to the introduction section

2. The role of recall bias

3. Before using the ecological model to describe your variables, briefly describe it.

4.

Results

1. About 35% of the sample size resident in the urban areas , is it reflect real population distribution across the country?

2. Indicate clearly if subgroup trends have been adjusted for other factors or are crude (unadjusted). It's crucial to clarify this restriction and highlight the corrected outcomes from multivariable models in the discussion if trends are not adjusted.

3.

Discussion

1. Make a stronger connection between the findings and public health priorities.

2.

References

Follow the journal guideline regarding references writing

1. There some references either very old or relatively old. So . it should be changed

**Do you want your identity to be public for this peer review?** For information about this choice, including consent withdrawal, please see our Privacy Policy

Reviewer #2: No

Reviewer #3: **Yes: ** Masood Abdulkareem Abdulrahman

---

## [Author Response · Author response to Decision Letter 2]

16 Nov 2025

We thank you for inviting us to submit our revised manuscript. We are grateful to you and the peer reviewers for their constructive and insightful comments, which significantly improved our work. We have incorporated that feedback into the revision. After revision and multiple rounds of editing, we have finalised the CLEAN COPY of our manuscript. We have also included the TRACK CHANGE copy and the point-by-point responses to the feedback.

---

## [Editor Report · Decision Letter 2]

27 Nov 2025

Dear Dr.  Singh,

Thank you for submitting your manuscript to PLOS ONE. After careful consideration, we feel that it has merit but does not fully meet PLOS ONE’s publication criteria as it currently stands. Therefore, we invite you to submit a revised version of the manuscript that addresses the points raised during the review process.

We look forward to receiving your revised manuscript.

Kind regards,

Frank Kyei-Arthur, Ph.D.

Academic Editor

PLOS ONE

Journal Requirements:

Additional Editor Comments:

The authors have satisfactorily responded to all reviewers’ comments. However, there are two minor corrections to be made below:

In Supplementary Table S3, there is an issue with the household size categories: they are presented as 1-Mar, 4-Mya, and Jun-38. Please correct the categories to 1-3, 4-5, and 6-38.Supplementary Table S4 has the same issue regarding the household size categories: they are presented as 1-Mar, 4-Mya, and Jun-38. Please correct the categories to 1-3, 4-5, and 6-38.

---

## [Author Response · Author response to Decision Letter 3]

28 Nov 2025

Dear Editor,

Thank you for providing us with the opportunity to make a minor revision to our supplementary table. We have made the necessary corrections in the supplementary tables.

regards

---

## [Editor Report · Decision Letter 3]

2 Dec 2025

Commercial milk formula feeding among children under two years in Nepal: Trends and determinants from four Nepal Demographic and Health Surveys (2006-2022)

PONE-D-24-48446R3

Dear Dr. Singh,

We’re pleased to inform you that your manuscript has been judged scientifically suitable for publication and will be formally accepted for publication once it meets all outstanding technical requirements.

Kind regards,

Frank Kyei-Arthur, Ph.D.

Academic Editor

PLOS ONE

Additional Editor Comments (optional):

The authors have addressed all my earlier comments, and the manuscript is suitable for publication. 
---

## [Editor Report · Acceptance letter]

PONE-D-24-48446R3

PLOS One

Dear Dr. Singh,

I'm pleased to inform you that your manuscript has been deemed suitable for publication in PLOS One. Congratulations! Your manuscript is now being handed over to our production team.

Kind regards,

on behalf of

Dr. Frank Kyei-Arthur

Academic Editor

PLOS One